# Chemical, Physical and Biological Triggers of Evolutionary Conserved Bcl-xL-Mediated Apoptosis

**DOI:** 10.3390/cancers12061694

**Published:** 2020-06-25

**Authors:** Aleksandr Ianevski, Evgeny Kulesskiy, Klara Krpina, Guofeng Lou, Yahyah Aman, Andrii Bugai, Koit Aasumets, Yevhen Akimov, Daria Bulanova, Kiira Gildemann, Albert F. Arutyunyan, Olga Yu. Susova, Alexei L. Zhuze, Ping Ji, Wei Wang, Toril Holien, Marit Bugge, Eva Zusinaite, Valentyn Oksenych, Hilde Lysvand, Joachim M. Gerhold, Magnar Bjørås, Pål Johansen, Anders Waage, Caroline A. Heckman, Evandro F. Fang, Denis E. Kainov

**Affiliations:** 1Department of Clinical and Molecular Medicine, Norwegian University of Science and Technology, 7028 Trondheim, Norway; aleksandr.ianevski@ntnu.no (A.I.); klarak@stud.ntnu.no (K.K.); ping.ji@ntnu.no (P.J.); wei.wang@ntnu.no (W.W.); toril.holien@ntnu.no (T.H.); marit.bugge@ntnu.no (M.B.); valentyn.oksenych@ntnu.no (V.O.); Hilde.Lysvand@ntnu.no (H.L.); magnar.bjoras@ntnu.no (M.B.); anders.waage@ntnu.no (A.W.); 2Institute for Molecular Medicine Finland (FIMM), University of Helsinki, 00014 Helsinki, Finland; Evgeny.Kulesskiy@helsinki.fi (E.K.); yevhen.akimov@helsinki.fi (Y.A.); daria.bulanova@helsinki.fi (D.B.); caroline.heckman@helsinki.fi (C.A.H.); 3Department of Clinical Molecular Biology, University of Oslo and Akershus University Hospital, 1478 Lørenskog, Norway; guofeng.lou@medisin.uio.no (G.L.); yahyah.aman@medisin.uio.no (Y.A.); fei.fang@medisin.uio.no (E.F.F.); 4Department of Biochemistry and Developmental Biology, University of Helsinki, 00014 Helsinki, Finland; andrii.bugai@helsinki.fi; 5Institute of Technology, University of Tartu, 50090 Tartu, Estonia; koitaasumets@gmail.com (K.A.); kiira.gildemann@ut.ee (K.G.); Eva.Zusinaite@ut.ee (E.Z.); gerhold@ut.ee (J.M.G.); 6Engelhardt Institute of Molecular Biology, Russian Academy of Sciences, 119991 Moscow, Russia; abosiy@yandex.ru (A.F.A.); zhuze@eimb.ru (A.L.Z.); 7Institute of Carcinogenesis, FSBI “N.N. Blokhin National Medical Research Center of Oncology”, The Ministry of Health of the Russian Federation, 119991 Moscow, Russia; susovaolga@gmail.com; 8Department of Hematology, St. Olav’s University Hospital, 7030 Trondheim, Norway; 9Department of Dermatology, University of Zurich, 8006 Zurich, Switzerland; pal.johansen@usz.ch

**Keywords:** apoptosis, Bcl-xL, oncolytics, navitoclax

## Abstract

Background: The evidence that pan-Bcl-2 or Bcl-xL-specific inhibitors prematurely kill virus-infected or RNA/DNA-transfected cells provides rationale for investigating these apoptotic inducers further. We hypothesized that not only invasive RNA or DNA (biological factors) but also DNA/RNA-damaging chemical or physical factors could trigger apoptosis that have been sensitized with pan-Bcl-2 or Bcl-xL-specific agents; Methods: We tested chemical and physical factors plus Bcl-xL-specific inhibitor A-1155463 in cells of various origins and the small roundworms (*C. elegans*); Results: We show that combination of a A-1155463 along with a DNA-damaging agent, 4-nitroquinoline-1-oxide (4NQO), prematurely kills cells of various origins as well as *C. elegans*. The synergistic effect is p53-dependent and associated with the release of Bad and Bax from Bcl-xL, which trigger mitochondrial outer membrane permeabilization. Furthermore, we found that combining Bcl-xL-specific inhibitors with various chemical compounds or physical insults also induced cell death; Conclusions: Thus, we were able to identify several biological, chemical and physical triggers of the evolutionarily conserved Bcl-xL-mediated apoptotic pathway, shedding light on strategies and targets for novel drug development.

## 1. Introduction

Apoptosis is a tightly regulated process that results in the death of cells with damaged or pathogenic DNA, RNA, and proteins [1,2]. When apoptosis is inhibited, cells that should otherwise be eliminated may persist and become malignant [3].

The B cell lymphoma 2 (Bcl-2) family of proteins are key players in apoptosis with opposing regulatory activity [4]. While Bcl-xL, Bcl-2 and Mcl-1 are anti-apoptotic, Bax, Bak and Bad are pro-apoptotic [5,6,7]. Interaction between pro- and anti-apoptotic proteins determine the fate of a cell. In particular, alteration of the interactions could lead to release of Bax and Bak, which form pores in the mitochondrial outer membrane, allowing cytochrome C release and activation of the caspase cascade [8].

Several chemical inhibitors have been developed to bind anti-apoptotic of Bcl-2 proteins and induce cancer cell death including Bcl-PROTACs [9]. These Bcl-2 inhibitors (Bcl2i) belong to several structurally distinct classes (Figure 1a). One major class includes the molecule ABT-737 and its derivatives, i.e., ABT-263 (navitoclax) and ABT-199 (venetoclax); while another class includes WEHI-539 and its derivatives, A-1331852 and A-1155463. Other classes of Bcl2i exist as well, and include molecules such as S63845 and S64315, S55746 and A1210477.

These Bcl2i have affinities to different Bcl-2 protein family members (Figure 1b). For example, ABT-199 has strong affinity to Bcl-2, WEHI-539, A-1331852 and A-1155463 are specific for Bcl-xL, S63845, S64315, S55746 and A1210477 have high affinity to Mcl-1, and ABT-263 binds Bcl-2, Bcl-xL and Bcl-w with similar affinity. Importantly, ABT-199 was approved, whereas ABT-263, S63845 and several other Bcl-2 inhibitors are currently in clinical trials against blood cancers and solid tumours [10,11,12,13] (NCT02920697). These drugs provide opportunities for treatment of hematologic and other types of malignancies, but also create new challenges associated with emerging drug resistance of cancer cells and toxicity for non-cancer cells (e.g., thrombocytopenia).

To enhance efficacy of treatment and combat genetically heterogeneous cancers, Bcl-2 inhibitors have been combined with other anticancer drugs, allowing for synergistic activity. (Figure 1c; bcl2icombi.info) [14,15,16,17]. Drug combinations have also been used to lower the dose of Bcl-2 inhibitors to overcome resistance and toxicity issues for non-malignant cells [9]. Dozens of the drug combinations have been reported to be active in vitro (cell culture, patient-derived cells or organoids) and in vivo (patient-derived xenograft mouse models). In addition, 109 combinations (excluding combinations with biological agents) have been tested in clinical trials. However, only ABT-199 in combination with azacytidine, decitabine or cytarabine has been approved for the treatment of acute myeloid leukaemia (AML). We noticed that many of these drug combinations also contain compounds that damage cellular macromolecules by targeting DNA replication, RNA transcription and decay, as well as protein signalling and degradation pathways. Here, we shed new light on the mechanisms of actions of such combinations. First, we show that a combination of 4NQO together with A-1155463, but not the drugs alone, leads to death in human malignant and non-malignant cells, as well as *C. elegans*. We found the synergistic effect of the drug combination to be p53 dependent. Moreover, we demonstrated that this process is associated with the release of Bad and Bax from Bcl-xL and activation of MOMP. Second, we demonstrated that several anticancer drugs (i.e., amsacrine, SN38, cisplatin, mitoxantrone, dactinomycin, dinaciclib, UCN-01, bortezomib, and S63845) trigger Bcl-xl-mediated apoptosis in non-malignant cells. Third, we show that combination of Bcl-xL specific inhibitors and UV radiation can also lead to cell death. We discuss the application of these results to cancer treatment.

## 2. Results

### 2.1. Toxicity of A-1155463-4NQO Combination in C. elegans

We demonstrated recently that A-1155463 prematurely killed virus-infected or RNA/DNA-transfected cells via Bcl-xL-mediated apoptosis [2,18,19]. We hypothesized that not only invasive RNA or DNA (biological factors) but also DNA/RNA-damaging chemical or physical factors could trigger apoptosis that have been sensitized with A-1155463. We tested DNA-damaging agent 4NQO [20] plus Bcl-xL-specific inhibitor A-1155463 in *C. elegans*. The worms treated with a combination of A-1155463-4NQO died faster than those treated with either A-1155463 or 4NQO alone (Figure 2a,b). Moreover, the worms treated with the drug combination exhibited defects in reproduction and development (Figure 2c). The Bliss synergy scores for adult development, L4 development and egg hatching were 19, 21 and 24, respectively. This synergy score can be interpreted as the average excess response due to drug interactions (i.e., 19, 21, and 24% of cell survival beyond expected additivity between single drugs). Treatment with A-1155463 or 4NQO alone did not affect these stages. Thus, combination of 4NQO with A-1155463 had a severe impact on the *C. elegans* lifespan, reproductive system and development.

### 2.2. Toxicity of A-1155463-4NQO Combination for Human, Monkey and Dog Cells

To determine the in vivo toxicity of A-1155463-4NQO in vitro, we used human non-malignant retinal pigment epithelium (RPE) cells. The cells died 3 h after treatment with A-1155463-4NQO, whereas majority of cells treated with either A-1155463 or 4NQO alone remained viable for 24 h as measured by real-time impedance assay (Figure 3a). Another cell viability assay, which quantified intracellular ATP, demonstrated that the effect of A-1155463-4NQO was synergistic (ZIP synergy score, 14 ± 3; Figure 3b). Similar results were obtained with 4NQO in combination with another Bcl-xL-specific inhibitor, A-133852 (ZIP synergy score, 17 ± 0; Figure 3c). ABT-263, a pan-Bcl-2 inhibitor, also showed synergy with 4NQO (ZIP synergy score, 8 ± 1; Figure 3c). In contrast, combinations of 4NQO with the Bcl-2- or Mcl-1-specific inhibitors did not show a synergy (ZIP synergy scores, 3 ± 1 and 2 ± 1, respectively; Figure 3c). These results suggested that the DNA-damaging agent combined with Bcl-xL-, but not Bcl-2- or Mcl1-specific inhibitors facilitated the death of human non-malignant cells.

We also tested the combinations of 4NQO-A-1155463 and 4NQO-A-1331852 in human cancer cell lines A549, H460 and Caco-2 (Figure 3d). In addition, we tested 4NQO-A-1155463 in a panel of patient-derived primary cell cultures (Figure 3e), monkey Vero-E6 and dog MDCK cells (Figure 3f). The combinations showed synergy in all tested cells, indicating that the DNA-damaging agent combined with Bcl-xL-specific inhibitor facilitated the death of monkey, dog, human non-malignant and malignant cells.

### 2.3. Concerted Action of 4NQO and A-1155463 Leads to Overexpression of p53, Release of Bad and Bax from Bcl-xL and Activation of MOMP

Immunoblot analysis of whole-cell extracts, nuclear and cytoplasm fractions showed that p53, a key regulator of DNA-damage response and Bcl-xL-dependent apoptosis, was over-expressed and accumulated in the nucleus of RPE cells after 2 h in response to 4NQO treatment (both as single agent and in combination; Figure 4a). Confocal microscopy also confirmed this observation (Figure 4b,c).

Immuno-precipitation of Bcl-xL-interacting partners showed that A-1155463 displaced Bad and Bax from Bcl-xL in RPE cells after 2 h of treatment (Figure 4d). This is in agreement with previous observations that BAX localizes to the outer mitochondrial membrane and dimerizes with BAK to form the pores and induce MOMP upon treatment with a Bcl-xL inhibitor [19,21].

We monitored MOMP with membrane-permeant JC-1 dye. JC-1 dye exhibits potential-dependent accumulation in apoptotic mitochondria, indicated by a fluorescence emission shift from green to red. The treatment of RPE cells with 4NQO-A-1155463 for 4 h decreased the red/green fluorescence intensity ratio, when compared to non-, 4NQO- or A-1155463-treated cells (Figure 4e,f). These results indicate that the concerted action of a DNA-damaging agent and Bcl-xL inhibitor induce MOMP.

Of note, treatment of RPE cells for 2 h with A-1155463-4NQO combination, but not with compounds alone substantially affected general translation and phosphorylation of several proteins(Appendix A). By contrast, protein synthesis as well as expression levels of several apoptotic proteins were not affected by the combination (Appendix A). These effects could represent the immediate early responses to apoptosis.

### 2.4. p53 is Required for A-1155463-4NQO Synergy

To test the significance of p53 overexpression in A-1155463-4NQO synergy, we used malignant HCT116 TP53^−/−^ cells that lack p53 expression (Figure 5a). A-1155463-4NQO combination had substantially lower effect on viability, death and early apoptosis in TP53^−/−^ cells (Figure 5b–d). By contrast, A-1155463-4NQO killed malignant HCT116 TP53^+/+^ cells, at the same concentrations as for RPE cells. These results indicated that p53 was essential for A-1155463-4NQO synergy.

We also tested A-1155463-4NQO in a panel of cancer cell lines with varying degrees of p53 expression (Figure 5e). Cell lines included in this panel were A549, NCIH460 (lung), Caco-2, SW480, HT29, SW620 (large intestine), JJN3, ANBL6, U266, RPMI8226, KJON, and INA6 (haematopoietic and lymphoid tissue). The experiment revealed high synergism (synergy scores >7.5) for the drug combination in A549, NCIH460, Caco-2, SW480, SW620, RPMI8226, and INA6 cells. Moderate synergism (synergy scores between 5 and 7.5) was observed in ANBL6 and U266 cells which had S241Y and A161T mutations in p53. A-1155463-4NQO- showed no or low synergism (synergy scores < 5) in JJN3, KJON and INA6 cell lines, which have deletions in *TP53* gene as well as in HT29 cells with R273H mutation in p53. Thus, the mutations in TP53 could affect the A-1155463-4NQO synergy.

### 2.5. Chemical Agents Triggering p53-dependent Bcl-xL-Mediated Apoptosis

Next, we tested A-1155463, A-133852, ABT-199, and S63845 in combination with 39 DNA, RNA and protein damaging agents in RPE cells. Our results revealed that A-133852 and A-1155463 acted highly synergistically in combination with these damaging agents, indicating that Bcl-xL is a major player in the induction of apoptosis under chemical insults (Figure 6a,b). Importantly, combinations with synergy scores >7.5 induced expression of p53 after 2 h of treatment (Figure 6c). Interestingly, combinations of S63845 with Bcl-xL, but not Bcl-2 inhibitors were highly synergistic (cytotoxic) indicating that S63845 could damage Mcl-1 or another potential cellular target (Figure 6d).

Moreover, screening of A-1331852 with 527 approved and emerging investigational anticancer agents revealed 64 combinations with synergy scores >7.5 (Appendix A). The hit anticancer agents targeted mainly cellular DNA replication (i.e., amsacrine, SN38, cisplatin, mitoxantrone, etoposide, dactinomycin), RNA transcription (i.e., dinaciclib, THZ2, alvocidib, fludarabine), protein signalling (i.e., AMG-232, UCN-01, pictilisib, triciribine) or cytoskeleton (i.e., idasanutiin, indibulin, vinorelbine) (Figure 6e). Thus, these agents induced p53-dependent Bcl-xL-mediated apoptosis.

We further exploited Bcl-xL-mediated apoptosis for identification of chemical agents that damage cellular DNA, RNA or proteins. We tested a library consisting of 48 drugs commonly dispensed in Norway as well as a random library consisting of 50 safe-in-man broad-spectrum antiviral agents. We also tested monomeric bisbenzimidazole-pyrrole MB2Py(Ac), dimeric benzimidazole-pyrroles DB2Py(4) and DB2Py(5), as well as dimeric bisbenzimidazoles DBA(3), DBA(5), and DBA(7) on A-1155463-sensitized and non-sensitized RPE cells (Appendix A). These 6 molecules were developed to bind to the minor groove of DNA [22,23].

We found that combination with bortezomib had a Bliss synergy score >7.5 indicating that bortezomib impaired protein degradation, which could be associated with accumulation of damaged proteins, which triggered Bcl-xL-mediated apoptosis [24]. Several compounds, including birth-control drug 17α-ethynylestradiol, immunosuppressant cyclosporin, antiviral agent brincidofovir, as well as DNA-binding probe MB2Py(Ac), in combination with A-1155463 had moderate synergy (synergy scores between 5 and 7.5), indicating that they could also trigger Bcl-xL-mediated apoptosis (Figure 7a–c).

We also tested combinations of A-1155463 with S63845, MB2Py(Ac), camptothecin, bortezomib or 17α-ethynylestradiol in human cancer A549, and H460 as well as monkey Vero-E6 and dog MDCK cells (Figure 7d). Combinations of A-1155463 with S63845, camptothecin or bortezomib were as highly synergistic as in RPE cells (except for A-1155463-bortezomib combination in H460 cells). Thus, we found drugs that damage cellular macromolecules can trigger apoptosis in human and animal cells.

### 2.6. Physical Factors Triggering Bcl-xL-mediated Apoptosis

We exposed A-1155463-sensitized and non-sensitized RPE cells to UVB and UVC radiation, which introduces DNA lesions similar to 4NQO treatment [20]. After 24 h, cell viability was measured using a CTG assay. We observed that only 8 s of UVB or UVC exposure was able to kill A-1155463-sensitized cells (Bliss synergy score = 31.2; Figure 8a). Cell death was not observed in non-sensitized RPE cells after the same amount of UVB or UVC exposure (Bliss synergy score = 23.8; Figure 8b). Similar effect was observed in UVC-exposed A-1155463-sensitized cancer A549, H460 and Caco-2 cell lines (Figure 8c). This indicated that UV light can trigger Bcl-xL-mediated apoptosis in non-malignant and malignant human cells.

## 3. Discussion

We showed recently that pan-Bcl-2 inhibitor ABT-263 accelerated killing of influenza virus-infected mice. Moreover, non-malignant cells infected with viruses or transfected with viral RNA or plasmid DNA were sensitive to Bcl-xL-specific inhibitor A-1155463 [2,18,19]. This indicates that invasive RNA or DNA, i.e., biological factors, could trigger apoptosis, and that Bcl-xL could serve as ‘safety fuse’ controlling the process (Figure 9).

Here, we shown that DNA-damaging agent 4NQO can kill human, monkey and dog cells, as well as small roundworms (*C. elegans*), that have been sensitized with A-1155463. Cell and organismal death was dependent on the concentration of both agents. The synergistic effect of 4NQO-A-1155463 in non-malignant cells was p53-dependent and was associated with the release of Bad and Bax from Bcl-xL. Our results suggest that intracellular receptors sense damage toDNA and transmit this information to anti-apoptotic Bcl-xL via p53. Bcl-xL in turn releases pro-apoptotic Bax and Bad to trigger MOMP leading to cell death. Thus, when the level of DNA damage reaches critical level, Bcl-xL can trigger apoptosis (Figure 9).

Combinations of 4NQO with Bcl-2- or Mcl-1-specific inhibitors did not show a synergy, suggesting that Bcl-xL has functional specialization in cells. Moreover, our results indicate that the Bcl-xL-mediated apoptotic pathway is evolutionary conserved (from *C. elegans* to *H. sapiens*). In addition, we identified other chemical triggers of Bcl-xL-mediated apoptosis. We showed that anticancer agents targeting DNA replication (i.e., amsacrine, SN38, cisplatin, mitoxantrone, etoposide, dactinomycin), RNA transcription (i.e., dinaciclib, THZ2, alvocidib, fludarabine), protein signalling (i.e., AMG-232, UCN-01, pictilisib, triciribine), protein quality control (i.e., bortezomib), or cytoskeleton (i.e., idasanutib, indibulin, vinoirelbine) killed cells sensitized with a Bcl-xL-specific inhibitor. We also identified physical triggers of Bcl-xL-mediated apoptosis. We observed that only 8s of UVB or UVC exposure can kill cells that have been sensitized with A-1155463, while non-sensitized cells are spared. This indicates that UV light triggers Bcl-xL-mediated apoptosis in A-1155463-sensitized cells.

Our results indicate that systemic treatment with combinations including pan-Bcl-2 or Bcl-xL-specific inhibitors (e.g., ABT-263 and A-1155463) could be harmful for cancer patients (similarly to round worms), because patients are constantly exposed to radiation, chemicals, and viral infections. Similarly, treatment with pan-Bcl-2 or Bcl-xL-specific inhibitors could be harmful for patients with viral diseases (similarly to mice) [19]. Thus, we identified adverse effects of pan-Bcl-2 or Bcl-xL-specific inhibitors.

These results may explain why clinical trials with 31 drug combinations containing pan-Bcl-2 inhibitor have been terminated, withdrawn or suspended. These include trials with ABT-263 plus bendamustine and rituximab in patients with relapsed diffuse large B cell lymphoma or ABT-263 plus abiraterone acetate with or without hydroxychloroquine in patients with progressive metastatic castrate refractory prostate cancer (NCT02471391, NCT01423539). 

Locally induced Bcl-xL-mediated apoptosis, however, could have a clinical potential. For example, skin cancers could be treated with local Bcl-xL-specific inhibitor plus UV radiation, oncolytic virus infection or anticancer medication. Recent studies support this idea by showing, for example, that combination of radiotherapy with pan-Bcl-2 inhibitors ABT-737 and ABT-263 can kill breast and small cell lung cancer cells in vitro (Appendix A) [25,26]. Moreover, ABT-737 sensitized chronic lymphocytic leukaemia cells to reovirus and vesicular stomatitis oncolysis [27]. In this respect, antibody-Bcl-xL inhibitor conjugates or Bcl-xL PROTACs can be used to achieve tissue selectivity and targeted delivery of the compounds.

## 4. Materials and Methods 

### 4.1. Website

We reviewed developmental status of Bcl-2 inhibitors and their combinations with other anticancer therapeutics. We summarized the information in freely accessible database (https://bcl2icombi.info). The drug annotations were obtained from PubChem, DrugBank, DrugCentral, PubMed and clinicaltrials.gov databases [28,29,30]. The database summarizes activities and developmental stages of the drug combinations and allows their interactive exploration. A feedback form is available on the website. The database will be updated upon request or as soon as a new drug combination is reported.

### 4.2. Compounds

ABT-199 (CAS: 1257044-40-8), A-1331852 (CAS: 1430844-80-6), A-1155463 (CAS: 1235034-55-5), and S63845 (CAS: 1799633-27-4) were purchased from Selleck Chemicals (Houston, TX, USA). 4NQO (CAS: 56-57-5) was from Merck Life Science (Espoo, Finland). Dimeric bisbenzimidazoles DBA(3), DBA(5), DBA(7), monomeric MB2Py(Ac), dimeric DB2Py(4) and DB2Py(5) benzimidazole-pyrroles were synthesized as described previously [22,23]. A library of 527 approved and emerging investigational oncology drugs were from the collection of the Institute of Molecular Medicine Finland, FIMM (www.fimm.fi/en/services/technology-centre/htb/equipment-and-libraries/chemical-libraries). A library of 48 drugs commonly dispensed in Norway was assembled based on Norwegian Prescription Database (www.norpd.no). Appendix A lists these compounds, their suppliers and catalogue numbers. A library of 50 safe-in-man broad-spectrum antivirals was published previously [31]. To obtain 10 mM stock solutions, compounds were dissolved in dimethyl sulfoxide (DMSO, Sigma-Aldrich, Steinheim, Germany) or milli-Q water. The solutions were stored at −80 °C until use.

### 4.3. Cells

Human telomerase reverse transcriptase-immortalized retinal pigment epithelial (RPE, ATCC) and epithelial colorectal adenocarcinoma (Caco-2, ATCC) cells were grown in DMEM-F12 medium supplemented with 100 U/mL penicillin/streptomycin (Pen/Strep), 2 mM L-glutamine, 10% FBS, and 0.25% sodium bicarbonate (Sigma-Aldrich, St. Louis, MO, USA). Human large-cell lung carcinoma NCI-H460 cells were grown in RPMI medium supplied with 10% FBS and Pen-Strep. Human adenocarcinoma alveolar basal epithelial cells (A549, ATCC) were grown in DMEM medium supplied with 10% FBS and Pen-Strep. Human colon cancer cell lines HCT116 *TP53^+/+^* and HCT116 *TP53^−/−^* were grown in McCoy’s 5A Medium (Sigma, M9309) supplemented with 10% FBS and Pen/Strep. The cell lines were maintained at 37 °C with 5% CO_2_. Human colon cancer cell lines HT29 (ATCC, catalog no. HTB-38), SW620 (ATCC, catalog no. CCL-227) and SW480 (ATCC, catalog no. CCL-228) were cultivated in RPMI 1640 supplemented with 10% FBS, 2 mM glutamine and 0.05% gensumycin.

The human multiple myeloma cell lines INA-6, ANBL-6, and JJN3 were kind gifts from Dr M. Gramatzki (University of Erlangen-Nurnberg, Erlangen, Germany), Dr D. Jelinek (Mayo Clinic, Rochester, MN, USA), and Dr. J. Ball (University of Birmingham, UK), respectively, whereas RPMI-8226 and U266 were from ATCC (Rockville, MD, USA), and KJON was established in our laboratory [32,33,34,35]. INA-6, ANBL-6, and JJN3 were grown in RPMI supplemented with 10% FBS and 2 mM L-glutamine. RPMI-8226 and U266 were grown in RPMI supplemented with 2 mM glutamine, and 20% or 15% FBS, respectively. KJON cells were grown in RPMI supplemented with 10% human serum (HS) (Department of Immunology and Transfusion Medicine, St. Olav’s University Hospital, Trondheim, Norway) and 2 mM L-glutamine. One ng/mL recombinant human interleukin (IL)-6 (Gibco, Thermo Fisher Scientific, Waltham, MA, USA), was added to the media of the IL-6 dependent myeloma cell lines INA-6, ANBL-6 and KJON. The cell lines were maintained at 37 °C with 5% CO_2_.

Bone marrow aspirates (n = 6) were obtained from patients with AML (n = 6) after informed consent, using protocols approved by a local institutional review board of Helsinki University Hospital and Comprehensive Cancer Center and in accordance with the Declaration of Helsinki. Mononuclear cells (MNCs) were isolated by density gradient separation (Ficoll-Paque PREMIUM; GE Healthcare, Little Chalfont, Buckinghamshire, UK) and immediately analyzed or vitally frozen for later use. Cells were maintained in mononuclear cell medium (MCM; Promocell, Heidelberg, Germany) or in a 25% HS-5 CM plus 75% RPMI 1640 medium mix. Cell viability was measured using the CellTiter-Glo (CTG) reagent (Promega, Madison, WI, USA), according to the manufacturer’s instructions, with a PHERAstar FS plate reader (BMG LABTECH, Ortenberg, Germany). The study protocols were approved by an ethical committee from Helsinki University Hospital (codes 239/13/03/00/2010 and 303/13/03/01/2011).

### 4.4. C. elegans Maintenance, Lifespan and Toxicity Assays

Standard *C. elegans* strain maintenance procedures were followed in all experiments [36,37,38,39]. Nematode rearing temperature was kept at 20 °C, unless noted otherwise. N2: wild type Bristol isolate was obtained from the Caenorhabditis Genetics Center (CGC).

For lifespan experiments, gravid adult worms were placed on NGM plates containing either A-1155463 (10 µM), 4NQO (10 µM), combination A-1155463 (10 µM) + 4NQO (10 µM) or vehicle control and seeded with OP50 to lay eggs. Progeny were grown at 20 °C through the L4 larval stage and then transferred to fresh plates in groups of 30-35 worms per plate for a total of 100 individuals per experimental condition. Animals were transferred to fresh plates every 2–4 days thereafter and examined every other day for touch-provoked movement and pharyngeal pumping, until death. Worms that died owing to internally hatched eggs, an extruded gonad or desiccation due to crawling on the edge of the plates were censored and incorporated as such into the data set. Survival curves were created using the product-limit method of Kaplan and Meier. The log-rank (Mantel–Cox) test was used for statistical analysis.

A series of toxicity experiments, including fecundity, egg hatching, larval development, were conducted using N2 Bristol isolate *C. elegans*, which were cultivated as previously described [36] and maintained at 20 °C. Briefly, for the toxicity assay animals were initially synchronised by bleaching gravid adults (adult day 1-4) to extract the eggs. Eggs were placed on nematode growth medium (NGM) plates seeded with *Escherichia coli* (OP50). L4 larvae were subsequently transferred onto fresh OP50-seeded NGM plates and allowed to grow to adulthood. Ten adult day 1 worms (n = 30–50/experimental condition) were transferred onto assay NGM plates with OP50 containing either A-1155463 (1, 10, 100 µM), 4NQO (1, 10, 100 µM), combination A-1155463 (10 µM) plus 4NQO (2, 10, 20 µM) or vehicle control. 

The adult worms were allowed to lay eggs for 3 h and were subsequently removed from the plate. The number of eggs laid was quantified as a measure of the reproductive capacity of worms. The following day the number of unhatched eggs and L1 larvae were counted in order to determine the efficiency of egg hatching. Development to L4 larvae was assessed 36 h later, as a measure of larval growth. Finally, growth of L4 larvae to adulthood was quantified after 16 h after the larval stage. 

The toxicity assay was conducted at 20 °C on 10 mL NGM plates seeded with 100 µL OP50 from an overnight culture. Drug compounds and vehicle solvents were dissolved in a total volume of 200 µL, sufficient to cover the entire surface of the plate, and were dried at room temperature for 1–2 h prior to the transfer of worms. Each chemical concentration was tested 3–5 times. Statistical analysis was conducted using One-Way ANOVA followed by Tukey’s multiple comparison test.

### 4.5. Real-time Iimpedance Assay

RPE cells were grown at 37 °C in 16-well E-Plates to 90% confluency. The plates were installed with golden electrodes at the bottom of the wells and a weak electrical current was constantly applied to the cell medium. The changes in the cell adherence indexes (CI) were monitored by the xCELLigence real-time drug cytotoxicity system (ACEA Biosciences, San Diego, CA, USA) as described previously [40]. When cells reached 90% confluency, 1 µM 4NQO, 1 µM A-1155463 or their combination were added. Control cells were treated with 0.01% of DMSO. CI were normalized and monitored for another 24 h.

### 4.6. Cell Viability Assay

Approximately, 4 × 10^4^ RPE or HCT116 cells per well of 96-well plate were treated with A-1155463, 4NQO or both compounds. Control cells were treated with 0.01% of DMSO. After 24 or 48 h, respectively, the viability of cells was measured using the Cell Titer Glo assay (CTG, Promega). Luminescence was measured using Victor X3 (PerkinElmer, Akron, Ohio, USA) or Synergy Mx plate readers (BioTeck, Bad Friedrichshall, Germany).

### 4.7. Cell Toxicity Assay

Approximately 4 × 10^4^ HCT116, JJN3, ANBL6, U266, RPMI8226, KJON, and INA6 cells were seeded per well on a 96-well plate. Cells were treated with 1 µM 4NQO, 1 µM A-1155463 or combination of both compounds. Control cells were treated with 0.01% of DMSO. After 24 h, the cell death was detected using the Cell Toxicity Assay (CTxG, Promega). Fluorescence was measured using a Victor X3 plate reader.

### 4.8. Early Apoptosis Assay

Approximately 4 × 10^4^ RPE, HCT116 *TP53^+/+^* or HCT116 *TP53^−/−^* cells were seeded per well on a 96-well plate, and treated with 1 µM 4NQO, 1 µM A-1155463, or both compounds. Control cells were treated with 0.01% of DMSO. Activation of apoptosis was assessed using RealTime-Glo Annexin V Apoptosis and Necrosis Assay (Promega). Luminescence was measured using Victor X3 plate reader. 

### 4.9. Apoptosis Arrays

Approximately 1 × 10^6^ RPE cells were seeded per well on a 6-well plate, and treated with 1 µM 4NQO, 1 µM A-1155463, or both compounds. Control cells were treated with 0.01% of DMSO. After 2 h, relative levels of apoptosis-related proteins were determined using proteome profiler human apoptosis array kit, respectively, as described in the manuals (R&D Systems, Minneapolis, MN, USA). Membranes were scanned using Odyssey system (Li-Cor, Lincoln, NE, USA).

### 4.10. Metabolic Labelling of Cellular RNA and Proteins

To label newly synthesized RNA or proteins, approximately 1 × 10^5^ RPE cells were seeded per well on a 12-well plate, and treated with 1 µM 4NQO, 1 µM A-1155463, or both compounds dissolved in 500 μL cell growth medium. Control cells were treated with 0.01% of DMSO. The medium was supplemented with 3 μL [alpha-P32]UTP (9.25 MBq, 250 μCi in 25 μL). Cells were incubated for 2 h at 37 °C and washed twice with PBS. Total RNA was isolated using and RNeasy Plus extraction kit (Qiagen, Hilden, Germany). RNA was separated on a 1% agarose gel. Total RNA was detected using ethidium bromide. The gel was dried and^32^P-labeled RNA was detected using autoradiography using Typhoon 9400 scanner (GE Healthcare).

In a parallel experiment, the compounds or DMSO were added to 500 μL cysteine- and methionine-free DMEM medium (Sigma-Aldrich, Germany) containing 10% BSA and 3 µL [^35^S] EasyTag Express protein labelling mix (7 mCi, 259 MBq, 1175 Ci/mmol in 632 mL; Perkin Elmer, Espoo, Finland). After 2 h of incubation at 37 °C cells were washed twice with PBS, lysed in 2× SDS-loading buffer and sonicated. Lysates were loaded and proteins were separated on a 10% SDS-polyacrylamide gel. ^35^S-labelled proteins were monitored using autoradiography and visualized using a Typhoon 9400 scanner (GE Healthcare).

### 4.11. UV Radiation Assay

RPE cells were exposed to UVC (λ = 254 nm) or to UVB (λ = 302 nm) using Hoefer UVC 500 Ultraviolet Crosslinker (20 J/cm^2^) or VM25/30/GX trans-illuminator as UV sources, respectively. 1 µM A-1155463 or 0.1% DMSO were added to the cell medium. After 24 h cell viability was measured using a CTG assay (Promega). The luminescence was read with a PHERAstar FS plate reader (BMG Labtech, Ortenberg, Germany).

### 4.12. Synergy Calculations

Cells were treated with increasing concentrations of a Bcl-2 inhibitor and other drug, as indicated. After 24 h cell viability was measured using CTG assay. To test whether the drug combinations act synergistically, observed responses were compared with expected combination responses. The expected responses were calculated based on Bliss reference model using SynergyFinder web-application [41]. For in vitro combinatorial experiments where the whole dose-response matrix was measured, a normalized Bliss reference model, i.e., Zero Interaction Potency (ZIP) model was utilized. Unpaired *t*-test *p*-value was calculated using stats R package v.3.6.3 (https://cran.r-project.org/bin/windows/base/old/3.6.3/), to test the null hypothesis that no difference exists between synergy score mean values of two groups. We considered drug combinations with the synergy scores >7.5 as highly synergistic, and drug combinations with the synergy scores between 5 and 7.5 as moderately synergistic.

### 4.13. Fractionation of RPE Cells

RPE cells were left untreated or treated with 1 μM A-1155463, 1 μM 4NQO, or both compounds for 2 h. The cells were washed with PBS and collected in cell lysis buffer consisting of 10 mM HEPES, pH 7.5, 10 mM KCl, 0.1 mM EDTA, 1 mM dithiothreitol, 0.5% Nonidet-40, and a protease inhibitor cocktail. The cells were placed on ice for 20 min and vortexed every 5 min, and then centrifuged at 12,000× *g* at 4 °C for 10 min. The supernatant was collected as a cytoplasmic fraction. The remaining cell pellets were washed three times with cell lysis buffer and resuspended in nuclear buffer consisting of 20 mM HEPES, pH 7.5, 400 mM NaCl, 1 mM EDTA, 1 mM dithiothreitol, and a protease inhibitor cocktail. The resuspended nuclear fraction was placed in ice for 30 min and centrifuged at 12,000× *g* at 4 °C for 15 min. The supernatant was collected as nuclear extracts. Protein concentrations for all cell fractions was determined using Bradford’s reagent (BioRad, Irvine, LA, USA).

### 4.14. Analysis of Mitochondrial Membrane Potential

RPE cells were left untreated or treated with 1 μM A-1155463, 1 μM 4NQO, or both compounds for 2 h. The mitochondrial potential was determined using JC-1 dye staining according to the manufacturer’s instructions (ThermoFisher Scientific). The cells were washed once with PBS and stained with 10 ug/mL of JC-1 dye in culture medium for 10 min. The images of JC-1 staining were captured using EVOS fluorescence microscope and the mitochondrial potential was indicated by the shift of the fluorescence intensity ratio of J-aggregate (535/590 nm) and monomers (485/530 nm).

### 4.15. Immuno-precipitation and Immuno-blotting

RPE cells were left untreated or treated with 1 μM A-1155463, 1 μM 4NQO, or both compounds for 2 h. Cells were lysed in buffer with 20 mM Tris-HCl, 0.5% NP-40, 150 mM NaCl, 1.5 mM MgCl_2_, 10 mM KCl, 10% Glycerol, 0.5 mM EDTA, pH 7.9 and protease inhibitor cocktail (Sigma). Bcl-xL-associated factors were immuno-precipitated using rabbit anti-Bcl-xL antibody (54H6, Cell Signalling Technology, Danvers, MA, USA) immobilized on the magnetic Protein G Dynabeads (Thermo Fisher Scientific). Normal rabbit IgG (sc-2025, Santa Cruz Biotechnology, Santa Cruz, CA, USA) was used to control immunoprecipitation with an equal volume of cell lysate. The immobilized protein complexes together with the appropriate whole-cell extract inputs were separated with SDS-PAGE followed by immunoblotting.

Proteins were transferred to nitrocellulose membrane (Sartorius). The membranes were blocked with 10% BSA (Santa Cruz Biotechnology) in TBS. Primary antibodies (rabbit anti-Bax, SC-493, 1:1000; rabbit anti-BAD, sc-493, 1:1000; rabbit anti-Bad, D24A9, 1:1000; mouse anti-p53, sc-126, 1:1000; mouse anti-GAPDH, sc-32233; 1:1000 (Santa Cruz Biotechnology); goat anti-p53, af1355, 1:1000 (RnD Systems) anti-Bcl-xL, 1:1000, 54H6) were diluted in TBS and added to the membranes. After overnight incubation at 4 °C, membranes were washed three times with TBS buffer containing 0.03% Tween 20 (Tween/TBS). Secondary antibodies conjugated to HRP (Santa Cruz Biotechnology, anti-mouse: m-IgGκ BP-HRP, sc-516102; anti-rabbit mouse anti-rabbit IgG-HRP, sc-2357) were added. Chemiluminescence was detected using the Western Lightning Chemiluminescence Reagent (Perkin Elmer) by ChemiDoc Imaging System (BioRad).

HCT116 *TP53^+/+^* and *TP53^−/−^* cells were left untreated or were treated with 1 μM A-1155463, 1 μM 4NQO, or both compounds. After 16 h, cells were lysed, and proteins were separated using 4–20% gradient SDS-PAGE. The proteins were transferred to a Hybond-LFP PVDF membrane. The membranes were blocked using 5% milk in Tris-buffered saline (TBS). p53 and GAPDH were detected using mouse anti-p53 and mouse anti-GAPDH antibodies as described above.

### 4.16. Confocal Microscopy

RPE cells were treated with 1 μM A-1155463, 1 μM 4NQO or both compounds. Control cells were treated with 0.1% DMSO. After 1 and 2 h cells were fixed using 3.3% formaldehyde in 10% FBS and PBS, pH 7.4. Cells were permeabilized using 10% FBS and 0.1% Triton X-100 in PBS. Bad, Bax, Bcl-xL, p53 and Tom40 were detected using rabbit anti-Bcl-xL (1:250; clone 54H6; Cell Signalling Technology), mouse anti-Bax (1:250; 2D2, sc-20067), rabbit anti-Bax (1:250; N-20, sc-493), rabbit anti-BAD (1:250; clone D24A9; Santa Cruz Biotechnology), mouse anti-p53 (1:250; Santa Cruz Biotechnology) and anti-Tom40 (1:250; Santa Cruz Biotechnology, sc-11414) antibodies were used. Secondary goat anti-mouse IgG (H + L) Alexa Fluor^®^ 488 and 568 (Thermo Scientific), goat anti-rabbit IgG (H + L) Alexa Fluor^®^ 488 and 568 (Thermo Scientific) antibodies were used. ProLong™ Gold Antifade Mountant (Invitrogen, Waltham, MA, USA) with DAPI was used for mounting of cells. Cells were imaged using Zeiss LSM710 confocal microscope (Zeiss, Oberkochen, Germany). 

## 5. Conclusions

In conclusion, we have identified chemical, physical and biological factors that induced Bcl-xL-mediated apoptosis. This could provide novel opportunities for discovery of therapeutic and adverse effects of medications as well as to identify conditions to prevent aging, treat non-communicable and communicable diseases, given that cells with high concentration of damaged DNA, RNA or proteins or with invasive DNA and RNA molecules would be selectively eliminated from an organism receiving a Bcl-xL-specific inhibitor.

## Figures and Tables

**Figure 1 cancers-12-01694-f001:**
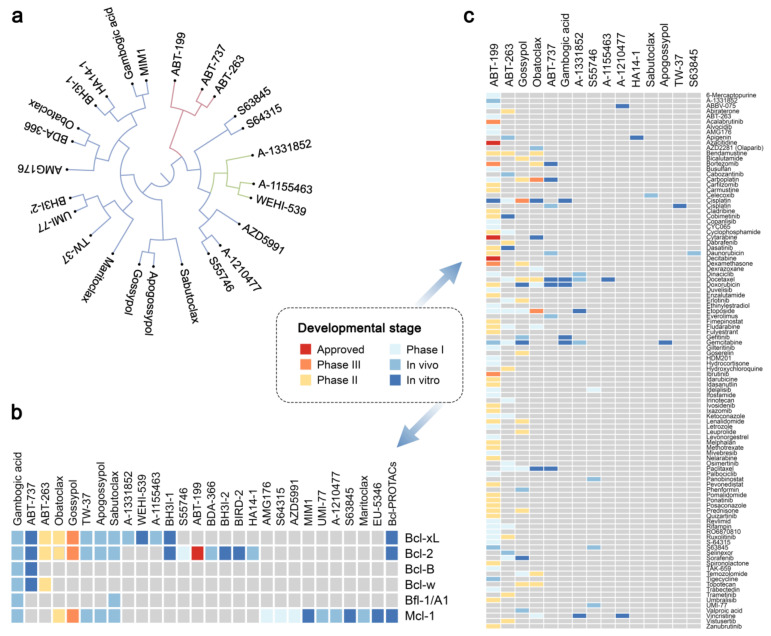
Bcl-2 protein family members and their combinations with other anticancer drugs. (**a**) Analysis of structural similarities between inhibitors of Bcl-2 proteins using C-SPADE. (**b**) Literature analysis of targets for inhibitors of Bcl-2 proteins. (**c**) Developmental status of drug combinations containing Bcl-2 inhibitors. For more information, please visit bcl2icombi.info.

**Figure 2 cancers-12-01694-f002:**
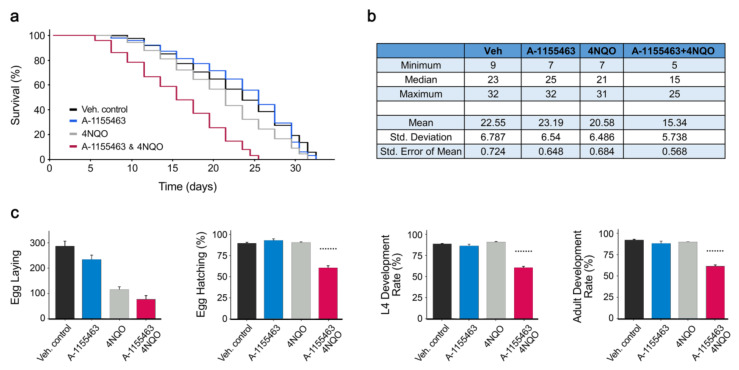
A combination of Bcl-xL- specific inhibitor A-1155463 and DNA-damaging agent 4NQO exhibits synergistic toxicity on lifespan, reproduction and development to the wild type roundworm *C. elegans*. (**a**) Kaplan-Meier survival curves and estimates of survival data (**b**) of worms challenged with 10 µM A-1155463, 10 µM 4NQO, drug combination or vehicle. The log-rank Mantel-Cox test was used for statistical analysis. (**c**) Toxicity detection for the different stages of *C. elegans* reproduction and development after the treatment with 10 µM A-1155463, 10 µM 4NQO or their combination (Mean ± SD). Dotted lines above combination bars represent the expected effect given by Bliss reference model. *******—expected effect of drug combination.

**Figure 3 cancers-12-01694-f003:**
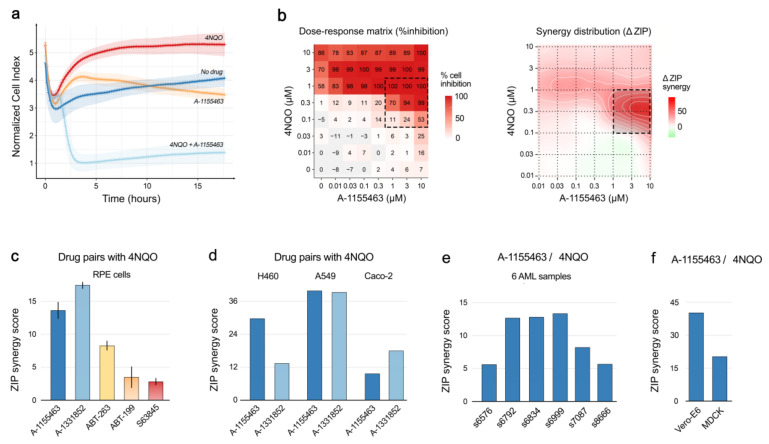
Combination of DNA-damaging agent 4NQO with Bcl-xL-, but not Bcl-2- or Mcl-1-specific inhibitors, exhibit synergistic toxicities on human non-malignant RPE, cancer A549, H460, Caco-2 and mononuclear cells (MNCs) isolated from AML patients as well as on monkey Vero-E6 and dog MDCK cells. (**a**) Real-time impedance traces for RPE cells exposed to 1 µM 4NQO, 1 µM A-1155463 or their combination. Control trace represents cells exposed to 0.1% DMSO (Mean ± SD; n = 8). (**b**) The interaction landscapes of A-1155463-4NQO combination. It represents the net combinational effects on viability of RPE cells, as measured with CTG assays. (**c**) Synergy scores of combinations of 4NQO and 5 Bcl-2 inhibitors on RPE cells (Mean ± SD; n = 3). Cells were treated with increasing concentrations of a Bcl-2 inhibitor and 4NQO. After 24 h cell viability was measured using the CTG assay. Synergy scores were quantified based on the ZIP model. (**d**) Synergy scores for combinations of 4NQO and 2 Bcl-xL inhibitors on human cancer A549, H460, Caco-2 cells. Cells were treated with increasing concentrations of a Bcl-xL inhibitor and 4NQO. After 24 h cell viability was measured using the CTG assay. Synergy scores were quantified based on the ZIP model. (**e**) Synergy scores for 4NQO- A-1155463 combination on MNCs. Cells were treated with increasing concentrations of a A-1155463, 4NQO or both agents. After 24 h cell viability was measured using the CTG assay. Synergy scores were quantified based on the ZIP model. (**f**) Synergy scores for 4NQO-A-1155463 combination for Vero-E6 and MDCK cells. Cells were treated with increasing concentrations of a A-1155463, 4NQO or both agents. After 24 h cell viability was measured using the CTG assay. Synergy scores were quantified based on the ZIP model.

**Figure 4 cancers-12-01694-f004:**
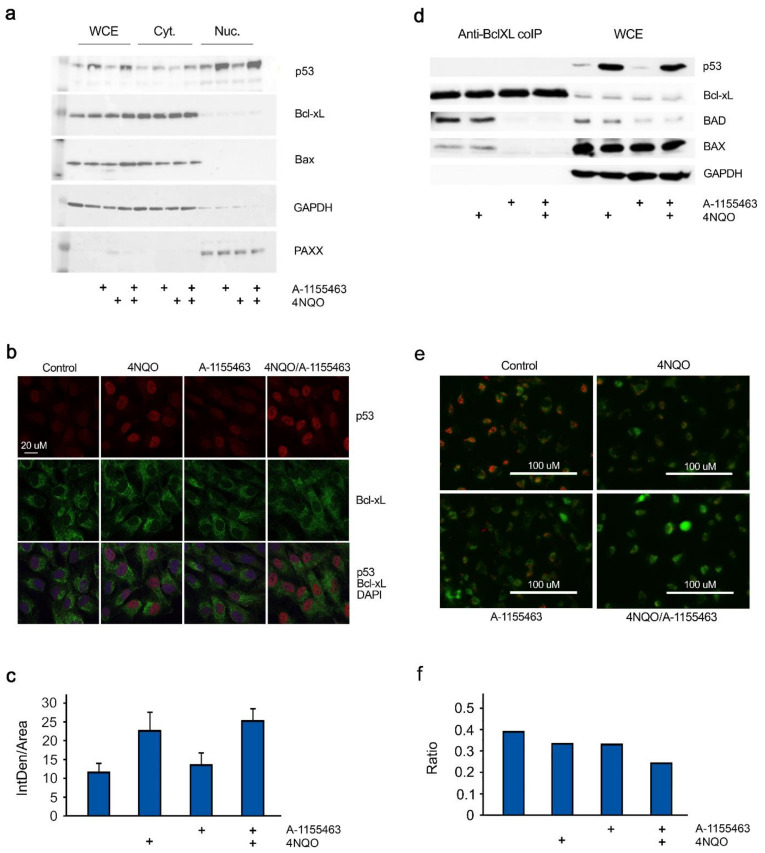
4NQO induces p53 expression, while A-1155463 triggers release of pro-apoptotic Bad and Bax from Bcl-xL to initiate MOMP in RPE cells. (**a**) RPE cells were treated with 1 μM 4NQO, 1 μM A-1155463 or their combination. Control cells were treated with 0.1% DMSO. Nuclear and cytoplasm fractions were prepared. P53, Bcl-xL, Bax, XLF, PAXX and GAPDH were analysed using Western blotting. (**b**) RPE cells were treated as for (a). Two hours after treatment cells were fixed. P53 and Bcl-xL were stained with corresponding antibodies. Nuclei were stained with DAPI. Cells were imaged using a confocal microscopy and representative images (n = 8) were selected. Scale bar, 20 μm. (**c**) Quantification of the red signal in the cells. (**d**) RPE cells were treated as for (a). Whole cell extracts (WCE) were obtained 2 h after treatment. Proteins were immuno-precipitated by anti-Bcl-xL antibody. P53, Bcl-xL, Bad, Bax, and GAPDH were analysed using Western blotting in WCE and immunoprecipitates. (**e**) RPE cells were treated as for (a). The mitochondrial potential was determined using JC-1 dye staining. The images of JC-1 staining were captured using fluorescence microscope and the mitochondrial potential was indicated by the shift of the fluorescence intensity ratio of J-aggregate (535/590 nm) and monomers (485/530 nm). Representative images (n = 3) were selected. Scale bar, 100 μm. (**f**) Quantification of the ratios between the green and red signals (indicative of mitochondrial membrane depolarization).

**Figure 5 cancers-12-01694-f005:**
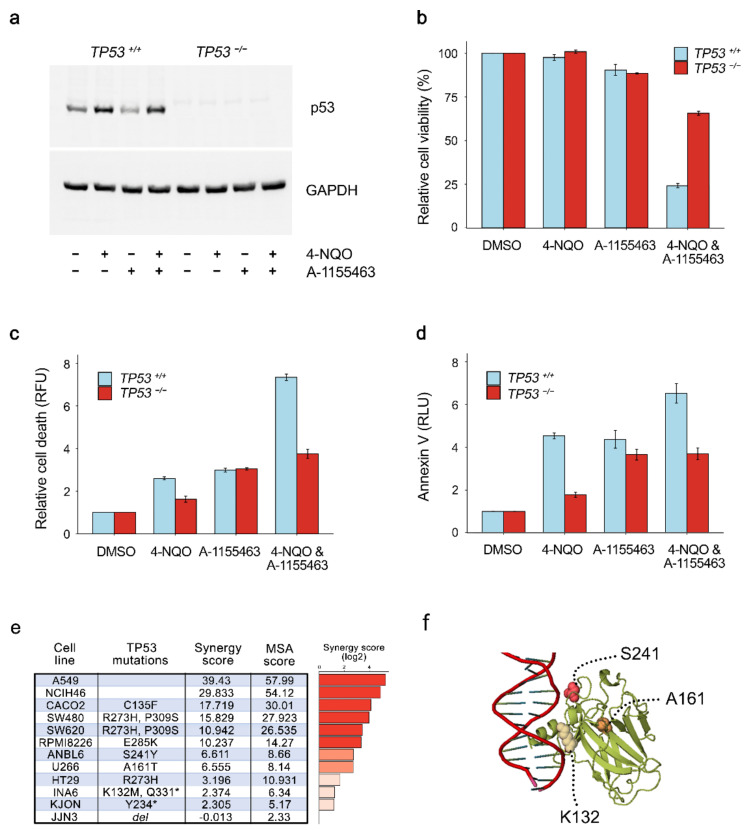
p53 is required for A-1155463-4NQO synergy. (**a**) HCT116 TP53^−/−^ and TP53^+/+^ cells were treated with 1 μM 4NQO, 1 μM A-1155463 or their combination. Control cells were treated with 0.1% DMSO. 2 h after treatment p53 and GAPDH were analyzed using western blotting of whole-cell extracts. (**b**) Cells were treated as for (a). Cell viability was measured by the CTG assay. Mean ± SD, n = 3. (**c**) Cells were treated as for (a). Cell death was measured by the CTxG assay. Mean ± SD, n = 3. RFU—relative fluorescence units. (**d**) Cells were treated as for (a). Apoptosis was measured by the Annexin V assay. Mean ± SD, n = 3. RLU—relative luminescence units. (**e**) Table showing p53 status (according to https://web.expasy.org/cellosaurus/) and sensitivity scores of A-1155463-4NQO combination for several cancer cell lines. MSA score—most synergistic area score. *-truncated version. (**f**) Ribbon representation of the p53-DNA complex focusing on residues at position 132, 161, and 241, which are mutated in some cell lines from (e) (PDB code 1TUP). These residues are shown in magenta.

**Figure 6 cancers-12-01694-f006:**
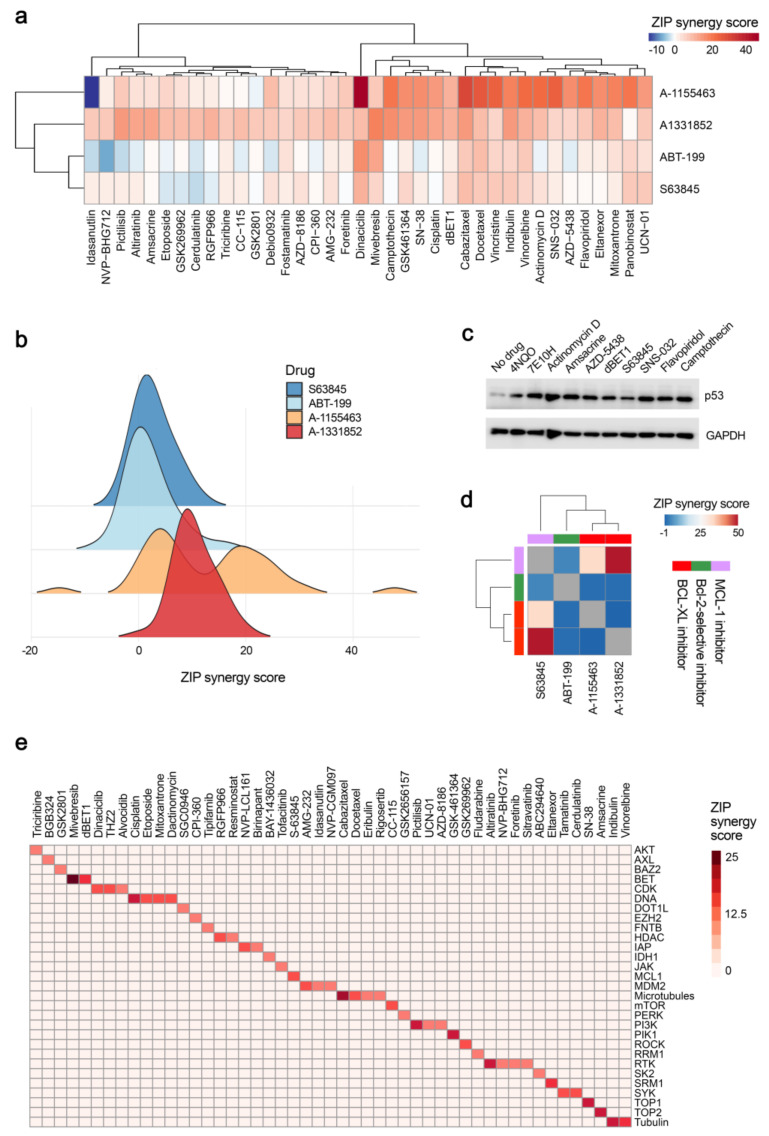
Identification of anticancer agents, which trigger Bcl-xL-mediated apoptosis. (**a**) Heatmap showing synergy scores for combinations of four Bcl-2 inhibitors and 39 anticancer agents. RPE cells were treated with 0, 10, 30, 100, 300, 1000, 3000 and 10,000 nM Bcl-2 inhibitors and anticancer agents. After 24 h cell viability was measured by the CTG assay. (**b**) Distribution of synergy scores based on (a). (**c**) Immunoblot analysis of p53 and GAPDH (loading control) expression levels after 2h of treatment of RPE cells with selected anticancer agents. (**d**) Heatmap showing synergy scores for combinations of four Bcl-2 inhibitors with each other. RPE cells were treated with 0, 10, 30, 100, 300, 1000, 3000 and 10,000 nM Bcl-2 inhibitors. After 24 h cell viability was measured by the CTG assay. (**e**) Cellular targets of anticancer agents which showed synergy with A-133852. RPE cells were treated with A-133852 and 527 anticancer agents as for panel (a). After 24 h cell viability was measured using the CTG assay. The compounds with synergy >7.5 were selected and plotted against their targets. ZIP method was used to calculate synergy scores for all panels.

**Figure 7 cancers-12-01694-f007:**
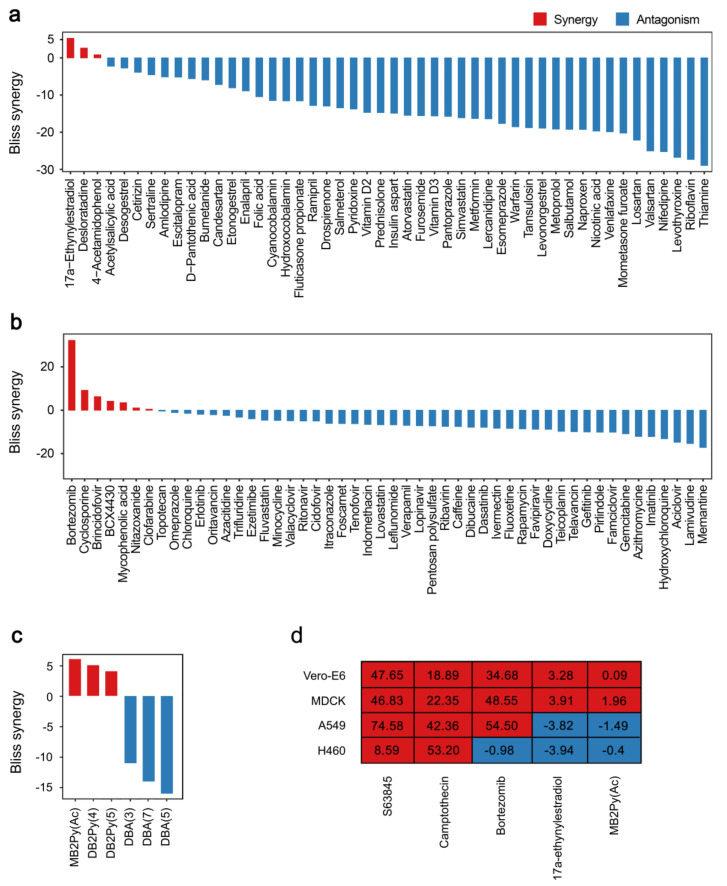
Identification of chemical agents, which trigger Bcl-xL-mediated apoptosis. (**a**) Bliss synergy scores for A-1155463 and 48 commonly dispensed drugs. A-1155463-sensitized (1 μM) and non-sensitized (0.1% DMSO) RPE cells were treated with 0, 0.1, 0.4, 1.2, 3.7, 11.1, 33.3, and 100 μM of drugs. After 24 h cell viability was measured by the CTG assay (n = 3). (**b**) Bliss synergy scores for A-1155463 and 50 safe-in-man broad-spectrum antiviral agents. A-1155463-sensitized (1 μM) and non-sensitized (0.1% DMSO) RPE cells were treated with 0, 0.04, 0.12, 0.37, 1.11, 3.33, 10, and 30 μM of antiviral agents. After 24 h cell viability was measured by the CTG assay (n = 3). (**c**) Bliss synergy scores for A-1155463 and DNA-binding agents. A-1155463-sensitized (1 μM) and non-sensitized (0.1% DMSO) RPE cells were treated with different concentrations of dimeric bisbenzimidazoles DBA(3), DBA(5) and DBA(7), monomeric bisbenzimidazole-pyrrole MB2Py(Ac) and dimeric benzimidazole-pyrroles DB2Py(4) and DB2Py(5). (**d**) Synergy scores of drug combinations for Vero-E6, MDCK, A549 and NCI-H460 cells. Cells were treated with increasing concentrations of a A-1155463, another agent or both agents. After 24 h cell viability was measured using the CTG assay. Synergy scores were quantified based on the ZIP model.

**Figure 8 cancers-12-01694-f008:**
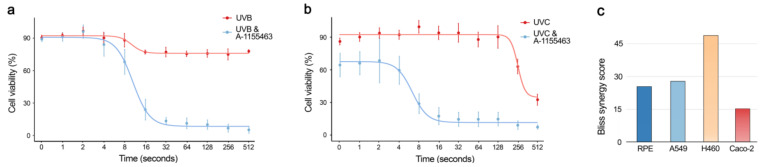
Identification of physical factors, which trigger Bcl-xL-mediated apoptosis. (**a**,**b**) RPE cells were exposed to UVB or UVC radiation for the indicated times and covered with medium containing 1 μM A-1155463 or 0.1% DMSO. After 24 h, viability of cells was measured using the CTG assay. Mean ± SD, n = 3. (**c**) Synergy scores of A-1155463-UVC combinations in 4 cell lines. Cells were treated with increasing A-1155463 concentrations and exposed UVC radiation for 0, 20, 40, 80, 160, and 240 sec. After 24 h cell viability was measured using the CTG assay. Synergy scores were quantified using Bliss model.

**Figure 9 cancers-12-01694-f009:**
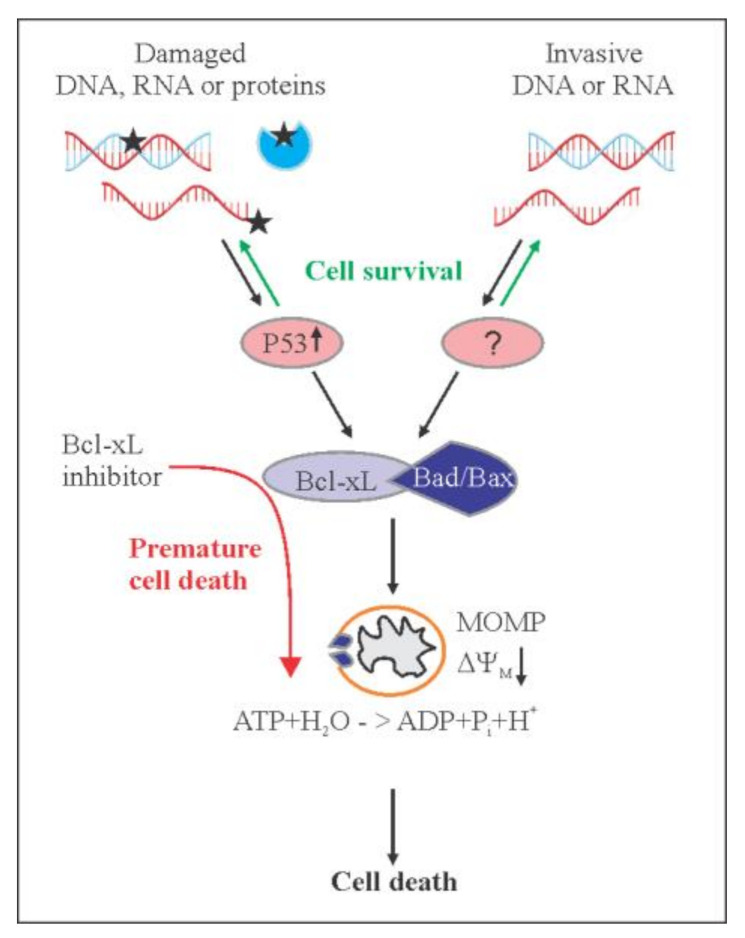
Schematic diagram showing how apoptosis could be initiated in response to chemical and physical stimuli or viruses, and how Bcl-xL-specific inhibitors could facilitate this process. Damage response (DR) factors and pattern recognition receptors (PRRs) recognize damaged molecules and invasive DNA or RNA, respectively. These proteins transduce signals to Bcl-xL via p53 or other protein(s). Pro-apoptotic Bax and Bad are released from anti-apoptotic Bcl-xL. When concentration of damaged or invasive molecules reach critical level, released Bax and Bad trigger mitochondrial outer membrane permeabilization (MoMP), which lead to inhibition of ATP synthesis, irreversible release of intermembrane space proteins, and subsequent caspase activation. This results in cell death. Alternatively, DR proteins and PRRs could mediate repair or degradation of damaged or invasive molecules, respectively, which would leave cells alive. Bcl-xL-specific inhibitors facilitate death of cells and, thus, impair repair or degradation of damaged or invasive factors.

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
