# Peer review of "Chemical, Physical and Biological Triggers of Evolutionary Conserved Bcl-xL-Mediated Apoptosis"

_cancers, 2020, doi:10.3390/cancers12061694_

Round 1

Reviewer 1 Report

Ianevski et al tested in the manuscript “Chemical, physical and biological triggers of evolutionary conserved Bcl-xL-mediated apoptosis” deals with the Bcl-xL-specific inhibitor A-1155463 that is tested alone and in combination in a wide cell line and primary cells containing screen either alone or in combination with a lot of other inhibitors. They include chemotherapeutics, proteasome inhibitors, BCL2- or MCL1 specific inhibitors. Combinations were tested for their ability to produce synergistic or additive effects. Furthermore, apoptosis-mediating factors were tested and analyzed on molecular level.

The work shows an impressive dataset and took for sure a lot of efforts.

As human non-malignant retinal pigment epithelium cells showed synergistic effects with DNA-damaging agent combined with Bcl-xL-inhibition, how sensemaking would therapies with it?

And how could it be explained that non-malignant retinal pigment epithelium cells die within 3 h upon treatment with A-1155463-4NQO whereas most other cell lines require at least 24 h? This substance seems in the end not to be a good inhibitor at all as it would cause a lot of side effects.

Typo: “transmitt“ line 318

Author Response

As human non-malignant retinal pigment epithelium cells showed synergistic effects with DNA-damaging agent combined with Bcl-xL-inhibition, how sensemaking would therapies with it?

Re: Our results indicat that systemic treatment with combinations including pan-Bcl-2 or Bcl-xL-specific inhibitors could be harmful for cancer patients (similarly to mice and round worms), because patients are constantly exposed to radiation, chemicals, and viral infections. Locally induced Bcl-xL-mediated apoptosis or combinations including Bcl-2-specific inhibitors (i.g. venetoclax), could be more appropriate for cancer treatment.

And how could it be explained that non-malignant retinal pigment epithelium cells die within 3 h upon treatment with A-1155463-4NQO whereas most other cell lines require at least 24 h?

Re: Like RPE, other cell lines died during the first hours after treatment with A-1155463-4NQO. A 24-hour time point was used in the study for consistency.

This substance seems in the end not to be a good inhibitor at all as it would cause a lot of side effects.

Re: Indeed, we identified adverse effects of A-1155463-4NQO and other combinational therapies, which include pan-Bcl-2 or Bcl-xL-specific inhibitors.

Reviewer 2 Report

The authors found that the combination of A-1155463 and other stimuli resulted in synergistic cytotoxicity via Bcl-xL in C. elegans, malignant and non-malignant cells. Cell death occurred through a p53-dependent pathway that triggered MOMP via the release of Bad and Bax from Bcl-xL. They identified several biological, chemical, physiological triggers of the evolutionarily conserved Bcl-xL-mediated apoptotic pathway, shedding light on strategies for novel drug development.

In the previous study, the authors reported that A-1155463 can be useful as an antiviral drug for influenza A virus because it induced premature death of IAV-infected cells. In this cancer study, their experiments covered many chemical agents as combinational partners of A-1155463. However, for the purpose of this paper, it is not clear whether the authors want to demonstrate the mechanisms by which known antitumor agent, ABT-263 with inhibitory activity on Bcl-2 and Bcl-xL, exerts cytotoxic effects on normal cells, leading to adverse events. Alternatively, do the authors want to show the disadvantage of using A-1155463 as an anticancer or antiviral agent? This needs to be clarfied.

There are other points to be clarified, as described below.

1.  This study demonstrated the importance of normal p53 function in the process of Bcl-xL inhibitor-mediated apoptosis. Does this mean that cancer cells with p53 mutation are less sensitive to A-1155463-4NQO treatment than non-malignant cells?

2. page 4, line 125: "Fig. 2c" would be "Fig. 3c". 

3. page 5, Figure 3a: Why did 4NQO treatment increase the cell viability?

4. page 5, line 160: "(Fig. 4c)" would be "(Fig. 4d)".

5. page 6, Figure 4a: This result shows expression of Bcl-xL in RPE cells. Is Bcl-xL also expressed in other malignant or non-malignant cells? Dose the combinational effect differ depending on the degree of Bcl-xL expression?

6. page 7, line 184: "(Fig. 4d,e)" would be "(Fig. 4e, f)".

7. page 7, line 196: "4NQO-A-1155463" can be referred to as "A-1155463-4NQO".

8. page 8, Figure 5d: RLU needs explanation. 4NQO or A-1155463 alone increased Annexin V-positive TP53+ cells to a level of 4.5 RLU. Combined, Annexin V-positive TP53+ cells increased to 7 RLU. Is this increase synergistic?

9. page 8, Figure 5e: MSA score may need explanation.

Author Response

In the previous study, the authors reported that A-1155463 can be useful as an antiviral drug for influenza A virus because it induced premature death of IAV-infected cells. In this cancer study, their experiments covered many chemical agents as combinational partners of A-1155463. However, for the purpose of this paper, it is not clear whether the authors want to demonstrate the mechanisms by which known antitumor agent, ABT-263 with inhibitory activity on Bcl-2 and Bcl-xL, exerts cytotoxic effects on normal cells, leading to adverse events. Alternatively, do the authors want to show the disadvantage of using A-1155463 as an anticancer or antiviral agent? This needs to be clarified.

Re: Many thanks to the reviewer for useful comments and suggestions for improving this manuscript. Our study shed new light on the MOA of  pan-Bcl-2 or Bcl-xL-specific inhibitors (e.g. ABT-263 and A-1155463). Our results indicate the disadvantage of A-1155463 and ABT-263 as anticancer or antiviral agents. We clarified the discussion section accordingly:

“Our results indicate that systemic treatment with combinations including pan-Bcl-2 or Bcl-xL-specific  inhibitors (e.g. ABT-263 and A-1155463) could be harmful for cancer patients (similarly to round worms), because patients are constantly exposed to radiation, chemicals, and viral infections. Similarly, treatment with pan-Bcl-2 or Bcl-xL-specific inhibitors could be harmful for patients with viral diseases (similarly to mice; Kakkola et al., 2013). Thus, we identified adverse effects of pan-Bcl-2 or Bcl-xL-specific inhibitors.”

There are other points to be clarified, as described below.

  1. This study demonstrated the importance of normal p53 function in the process of Bcl-xL inhibitor-mediated apoptosis. Does this mean that cancer cells with p53 mutation are less sensitive to A-1155463-4NQO treatment than non-malignant cells?

Re: Yes. A-1155463-4NQO showed no or low synergism (synergy scores < 5) in JJN3, KJON and INA6 cell lines, which have deletions in TP53 gene, as well as in HT29 cells with R273H mutation in p53 (Fig. 5e,f).

  1. page 4, line 125: "Fig. 2c" would be "Fig. 3c".

Re: Corrected to "(Fig. 3c)". Page 4, line 125.

  1. page 5, Figure 3a: Why did 4NQO treatment increase the cell viability?

Re: We do not know. DNA damage can, but does not always, induce cell death (Borges et al., 2009).

  1. page 5, line 160: "(Fig. 4c)" would be "(Fig. 4d)".

Re: Corrected to "(Fig. 4d)". Page 5, line 160.

  1. page 6, Figure 4a: This result shows expression of Bcl-xL in RPE cells. Is Bcl-xL also expressed in other malignant or non-malignant cells? Does the combinational effect differ depending on the degree of Bcl-xL expression?

Re: The response to BH3 mimetics sometimes correlates with the relative expression levels of the pro-survival proteins of Bcl-2 family (Ashkenazi A. et al, Nat. Rev. Drug Discov. 2017, 16, 273-284). Therefore, the combinational effect could differ depending on the degree of Bcl-xL and other protein expression, involved into Bcl-xL-mediated apoptosis (please see attached file with heatmap).

  1. page 7, line 184: "(Fig. 4d,e)" would be "(Fig. 4e, f)".

Re: Corrected to "(Fig. 4e, f)". Page 7, line 184.

  1. page 7, line 196: "4NQO-A-1155463" can be referred to as "A-1155463-4NQO".

Re: Corrected to "A-1155463-4NQO". Page 7, line 184.

  1. page 8, Figure 5d: RLU needs explanation. 4NQO or A-1155463 alone increased Annexin V-positive TP53+ cells to a level of 4.5 RLU. Combined, Annexin V-positive TP53+ cells increased to 7 RLU. Is this increase synergistic?

Re: We now explained RLU and RFU in the legend to Fig. 5: “RFU – relative fluorescence units. RLU – relative luminescence units”. The increase in RLU was significant (p = 0.01, unpaired t-test).

  1. page 8, Figure 5e: MSA score may need explanation.

Re: We now explained MSA score in the legend to Fig. 5e: “MSA score – most synergistic area score”

Typo: “transmitt“ line 318

Re: Corrected to “transmit”, line 318.

Reviewer 3 Report

Why have the authors used 4NQO? would X- or Gamma-ray been more appropriate? Is 4-NQO causing the same level of DNA damage as radiation?

Author Response

We used 4NQO because it is well known DNA-damaging agent (Walker & Sridhar, 1975). One of our authors recently shed more light on the MOA of 4NQO (Bugai et al., Mol Cell, 2019). 4NQO induces DNA lesions which could be corrected by nucleotide excision repair (NER, Seo et al., 1999; Ide et al., 2001). NER also repairs UV-induced DNA photoproducts.
Ionizing radiation induces DNA breaks which are repaired by non-homologous end joining (NHEJ) or homologous recombination in mammalian cells (Kumar, Alt, Oksenych, 2014). In our study, we used several compounds which, like ionizing irradiation, cause DNA breaks in human cells. For example, actinomycin D causes DNA single and double strand breaks probably by poisoning the topoisomerase enzymes (Ross and Bradley, 1981). We showed that combination of a Act D along with A-1155463, prematurely kills RPE cells (Fig. 6A). Thus, DNA damages of different origines trigger Bcl-xL-mediated apoptosis.